# CodY: An Essential Transcriptional Regulator Involved in Environmental Stress Tolerance in Foodborne *Staphylococcus aureus* RMSA24

**DOI:** 10.3390/foods12173166

**Published:** 2023-08-23

**Authors:** Hao Pei, Chengfeng Zhu, Fang Shu, Zhengfei Lu, Hui Wang, Kai Ma, Jun Wang, Ranxiang Lan, Fei Shang, Ting Xue

**Affiliations:** 1School of Life Sciences, Anhui Agricultural University, Hefei 230036, China; m18963761180_2@163.com (H.P.); 17309611151@163.com (C.Z.); s18256501178@163.com (F.S.); lzf6663@163.com (Z.L.); wang28hui@163.com (H.W.); makaiunique@163.com (K.M.); 15395065833@163.com (J.W.); lrx17398375169@163.com (R.L.); shf@ahau.edu.cn (F.S.); 2Food Procession Research Institute, Anhui Agricultural University, Hefei 230036, China

**Keywords:** *Staphylococcus aureus*, raw milk, CodY, environmental stress

## Abstract

*Staphylococcus aureus* (*S. aureus*), as the main pathogen in milk and dairy products, usually causes intoxication with vomiting and various kinds of inflammation after entering the human body. CodY, an important transcriptional regulator in *S. aureus*, plays an important role in regulating metabolism, growth, and virulence. However, little is known about the role of CodY on environmental stress tolerance. In this research, we revealed the role of CodY in environmental stress tolerance in foodborne *S. aureus* RMSA24. *codY* mutation significantly reduced the tolerance of *S. aureus* to desiccation and oxidative, salt, and high-temperature stresses. However, *S. aureus* was more tolerant to low temperature stress due to mutation of *codY*. We found that the expressions of two important heat shock proteins—GroEL and DanJ—were significantly down-regulated in the mutant *codY*. This suggests that CodY may indirectly regulate the high- and low-temperature tolerance of *S. aureus* by regulating the expressions of *groEL* and *danJ*. This study reveals a new mechanism of environmental stress tolerance in *S. aureus* and provides new insights into controlling the contamination and harm caused by *S. aureus* in the food industry.

## 1. Introduction

*Staphylococcus aureus* (*S. aureus*), a common Gram-positive pathogenic bacterium, usually colonizes the nasal cavity, skin, or mucous membranes of the host (human and animal) without symptoms [1]. As the pathogen of a variety of animal and human diseases, *S. aureus* is ubiquitous in the human environment. About 30% of healthy people carry *S. aureus*, which greatly increases the probability of infection [2]. This pathogen, which infects the body through wounds and spreads through the blood, can secrete extracellular toxins that destroy host cells and tissues, such as panton-valentine leukocidin and hemolysin, causing local abscesses, sepsis, endocarditis, and osteomyelitis [3]. In addition, *S. aureus* produces enterotoxins, soluble protein molecules with emetic activity that cause diarrhea and abdominal pain and are a major cause of food poisoning [4].

*S. aureus* is widely present in milk or dairy products and can cause foodborne outbreaks and serious infections worldwide. Two traditional food processing methods, high-temperature processing and drying, are widely used to control *S. aureus* contamination [5]. As a common food processing method, pickling can effectively prevent food spoilage [6]. By increasing the salt concentration in food, the growth of foodborne pathogens can be effectively inhibited, thus extending the shelf life of food [7,8]. However, *S. aureus* can be grown under a variety of environmental conditions, such as at high temperatures and in the context of high salt concentrations [9]. *S. aureus* can survive in a dry environment and spread through air or materials [10]. Several studies have shown that *S. aureus* can survive for up to 90 days in some dry objects [11]. For some environments where sugars and certain amino acids are present, the survival time is even longer [12]. Previous studies have found that antioxidant enzymes KatA and AhpC play indispensable roles in the desiccation stress response of *S. aureus* [13]. Although it has been reported that many *S. aureus* strains are tolerant to desiccation-related diseases, the underlying molecular mechanisms remain largely unknown. In addition, these antioxidant enzymes (KatA and AhpC) are also involved in the oxidative stress response. The hydrogen peroxide regulator PerR protects against oxidative stress by controlling the expression of some antioxidant enzymes KatA, Tpx, and Bcp [14]. The low-molecular-weight thiol bacillithiol also plays an important role in the oxidative stress response [15]. *S. aureus* shows a strong tolerance to salt, which allows it to survive in pickled foods. It has been reported that *S. aureus* can survive in sodium concentrations up to 900 mM, which is dependent on the two-component system KdpE/D [16,17,18]. Under salt stress conditions, the sensor histidine kinase KdpD activates the response regulator KdpE, which in turn binds to the promoter of the operon kdpFABC to activate the expression of the KdP-atpase potassium transporter to counteract Na+ cytotoxicity in *S. aureus* cells by increasing cytoplasmic K^+^ concentrations [19,20,21]. Therefore, revealing the mechanisms underlying the tolerance of *S. aureus* to environmental stress is necessary to control contamination by *S. aureus*.

The environmental tolerance of *S. aureus* is dependent on the functions of various transcriptional regulators. SigB is an important transcriptional regulator present in many Gram-positive bacteria, and it is mainly involved in environmental stress response [22,23,24]. The function of SigB in *Listeria* and *Staphylococcus* has been well studied [25,26]. In *S. aureus*, SigB is involved in promoter recognition and RNA polymerase recruitment in *S*. *aureus* [27]. In foodborne *S. aureus*, the mutation of *sigB* results in a significantly reduction in tolerance to heat, desiccation, and oxidative stress [28]. Transcription regulator sarA is present in a variety of bacteria (*Bacillus*, *Staphylococcus*, and *Listeria*) and can regulate the expression of multiple target genes. These target genes are mainly involved in bacterial biofilm formation, virulence, metabolism, and stress. In *S. aureus,* SarA regulates the expression of about 120 genes [29,30]. In addition, SarA affects the oxidative stress resistance of *S. aureus* by directly controlling the expression of sod [31].

CodY was first discovered in *Bacillus subtilis* in 1995 and is thought to be an inhibitor of the *B. subtilis* dipeptide penetrase gene [32]. CodY is a transcription regulator that is widely present in bacteria and has been found in almost all Gram-positive bacteria, including *Bacillus*, *staphylococcus* and *Listeria* [33]. The CodY generally plays a negative regulatory role by competing with RNA polymerase for promoter binding or interfering with positive regulator binding [34,35]. In many Gram-positive bacteria, the global regulatory factor CodY is mainly involved in metabolic and virulence factor synthesis [36]. In *Bacillus subtilis*, CodY regulates the expression of multiple genes and operons [37,38]. CodY is activated by environmental signals such as branched chain amino acids and GTP to regulate the expression of metabolism and virulence genes in *Bacillus subtilis* [39,40,41]. In *Streptococcus thermophilus*, CodY responds to oxidative stress by regulating glutathione biosynthesis [42]. In foodborne *S. aureus* strains, CodY has not been reported. The role of CodY in environmental tolerance is also poorly understood.

Heat stress proteins (Hsps) are a group of proteins that are produced when organisms are subjected to heat stress (high or low temperatures) [43]. When cells are subjected to heat stress, heat stress proteins are activated and expressed, and they play a role in protecting cells and maintaining cell homeostasis [44]. GroEL and DanJ, as important Hsps in bacteria, play an important role in the correct translation, folding, unfolding, translocation, and degradation of proteins under heat stress conditions [45,46].

The *codY* insertion mutant (M-*codY*) of *S. aureus* RMSA24 was obtained in our previous study [47]. In this paper, we reported the effects of *codY* mutation on the growth and survival of foodborne *S. aureus* RMSA24 under various food processing conditions (harsh temperature, drying, salt, and H_2_O_2_). In the present paper, we report the role of CodY in environmental stress tolerance and provide a potential target gene for revealing the mechanism of environmental stress tolerance. Our study may provide a theoretical basis for improving food processing and storage methods to control food contamination by *S. aureus*.

## 2. Materials and Methods

### 2.1. Strains and Growth Condition

The *S. aureus* strain RMSA24 used in this study was derived from our previous study [48]. The *S. aureus* was cultured in TSB (Oxoid, Basingstoke, UK) at 37 °C. Antibiotics (Sangon Biotech, Shanghai, China) such as ampicillin (100 μg/mL), chloramphenicol (15 μg/mL), and erythromycin (5 μg/mL) were added to the TSB when needed. The strains and plasmids used in this study are listed in Table 1.

### 2.2. Construction of codY Complementary Mutant

The *codY*-inserted mutant (M-*codY*) of *S. aureus* RMSA24 originated from our previous study [47]. Primers *codY*-F/*codY*-R were used to amplify the complementary expression cassette of *codY* gene from the RMSA24 genome. *SacI*/*HindIII* (Thermo Fisher Scientific, Waltham, MA, USA) endonucleases were used to digest the plasmid pLI50. Then, the Cloning Mix Kit (Monad, Wuhan, China) was used to connect the complementary expression cassette and digested plasmid. The complementary plasmid plasmid was firstly transferred to RN4220 for modification and then transferred to M-*codY* to get *codY* complementary strain C-*codY* (Table 1). The pLI50 was also transferred into RMSA24 to exclude the effect of plasmids on bacteria. Polymerase chain reaction was used to verify C-*codY* with the primers check-pLI50-f/check-pLI50-r. All of the primers used in this study are listed in Table 2.

### 2.3. Growth Curve Determination

All the strains were first cultured in tryptone soybean broth for 14 h and then incubated at a ratio of 1:100 into 100 mL fresh TSB containing 15 µg/mL chloramphenicol and incubated for 24 h at 37 °C in a shaker. The bacterial suspensions were taken every two hours for OD_600_ determination using a spectrophotometer (Thermo 157 Scientific, Pittsburgh, PA, USA), and growth curves were drawn using GraphPad Prism 9 software. The assay was repeated in triplicate.

### 2.4. Heat Stress Assay

After the single colony of *S. aureus* was incubated in TSB for 14 h, 1 mL of the culture was placed in fresh TSB medium. Then, the cell suspension was incubated to the late-exponential phase for the following stress experiments. The cultures of three strains were grown at 63 °C for thirty minutes or sixty minutes, respectively. The bacterial suspensions before and after treatment were diluted in 10-fold gradient and then uniformly spread on a TSB agar plate for colony counting. The survival rate was calculated by X/X_0_, where X_0_ is the number of colonies present in the sample before treatment, and X is the number of bacteria present in the sample after heat treatment. Each assay was repeated three times.

### 2.5. Cold Stress Assay

The bacterial suspensions were injected into tryptone soybean broth at a ratio of 1:20 and incubated at −20 °C and −80 °C for 2 h. The survival rates of all strains were measured according to the above method. Each assay was repeated three times.

### 2.6. Desiccation Stress Assay

The drying stress assays were finished based on the previous method [48]. The 200 μL bacterial suspensions of WT, M-*codY*, and C-*codY* were added to a 48-well plate, respectively, and incubated under desiccation conditions (humidity: 25% ± 5%; water activity: 0.38075 ± 0.01176) for 4 d. The survival rates of all strains were measured according to the above method. The assay was repeated in triplicate.

### 2.7. H_2_O_2_ Stress Assay

To test the tolerance of WT, M-*codY*, and C-*codY* to oxidative stress, we made some modifications to the method of our previous study [28]. The bacterial suspension was washed twice with sterile PBS, after which the washed bacterial suspension was collected by centrifugation. Then, the experimental group was treated with PBS containing hydrogen peroxide for 30 or 60 min, and the control group was treated with PBS without hydrogen peroxide for 30 or 60 min. The survival rates of all strains were measured according to the above method. The assay was repeated in triplicate.

### 2.8. Salt Stress Assay

The salt stress assay method used was based on our previous research [50]. One milliliter of the cell suspension was added into one hundred milliliters of fresh tryptone soybean broth containing ten percent sodium chloride, or no sodium chloride, and cultured in a shaker for 24 h. The OD_600_ of bacterial suspension was measured using a spectrophotometer (Thermo 157 Scientific, Pittsburgh, PA, USA) every two hours, and the growth curve was drawn using GraphPad Prism 9. The assay was repeated three times.

### 2.9. Reverse Transcriptase Quantitative PCR Experiment

Reverse Transcriptase quantitative PCR (RT–qPCR) was adopted to detect the expression of *groEL* and *danJ* in WT, M-*codY*, and C-*codY*. The total RNA of *S. aureus* was extracted using a Spin Column Bacteria Total RNA Purification Kit (Sangon Biotech, Shanghai, China), and the purity and quantity of RNA were tested using the Thermo Scientific NanoDrop Lite Spectrophotometer. The synthesis of cDNA was finished using *EasyScript*^®^ One-Step gDNA Removal and cDNA Synthesis SuperMix (TransGen Biotech, Beijing, China). Housekeeping gene *hu* was used as reference gene. All the primers used are listed in Table 2. *TransStart*^®^ Tip Green qPCR SuperMix (TransGen Biotech, Beijing, China) was used to carry out qPCR. The assay was repeated in triplicate.

### 2.10. Statistical Analysis

Statistical analyses were carried out using GraphPad Prism 8.0 (GraphPad Software Inc., GraphPad Prism 8.0.1.244, San Diego, CA, USA). The differences between different groups were analyzed using Student’s *t*-test, and *p* values < 0.05 were considered significant.

## 3. Results

### 3.1. Mutation of codY Had No Significant Effect on RMSA24 Growth

The inserted mutation M-*codY* was derived from our previous study [47]. To investigate the function of *codY* in RMSA24, we constructed the complementation strain C-*codY.* All the mutants were verified via polymerase chain reaction using primers check-pLI50-f/ check-pLI50-r. As shown in Figure 1A, a PCR product about 0.25 kb was amplified from the plasmids of WT (lane 2), M-*codY* (lane 3), or pLI50 (lane 4). A PCR product close to 1 kb was amplified from the plasmid of C-*codY* (lane 2). To investigate whether *codY* regulates the growth of RMSA24, we injected WT, M-*codY*, and C-*codY* into TSB for shaking. Then, the OD_600_ were measured to establish growth curves (Figure 1B). As shown in Figure 1B, the mutation of *codY* had no obvious effect on the growth and reproduction of *S. aureus*.

### 3.2. CodY Positively Regulates the Desiccation Tolerance of RMSA24

Drying is a common stabilization and conservation method in dairy processing [51]. To analyze the function of *codY* in the drying tolerance of *S. aureus*, the three strains were dried for 4 days, and the survival rates were calculated and compared. We found that the mutation of *codY* significantly reduces drying tolerance in *S. aureus* RMSA24 (Figure 2). Our results also suggest that CodY positively regulates the drying resistance of *S. aureus*.

### 3.3. CodY Is Involved in Regulating the Temperature Stress Response of RMSA24

In order to investigate the function of *codY* in the high-temperature stress response of RMSA24, we subjected WT, M-*codY*, and C-*codY* to high-temperature (63 °C) stress and calculated the survival rates. We found that the mutation of *codY* clearly reduced the tolerance of RMSA24 to high-temperature stress, as shown in Figure 3. The extremely low survival rate of *M-codY* after high-temperature treatment suggests that CodY is likely to be an essential transcriptional regulator in RMSA24 under high-temperature stress.

Moreover, survival rate under low-temperature stress was also examined. We counted the colonies of the WT, M-*codY*, and C-*codY* bacterial suspensions before and after cold treatment. The survival rate of M-*codY* after low-temperature treatment (−20 °C and −80 °C) was significantly increased compared to WT and C-*codY* (Figure 3C,D).

To further explore the mechanism by which *codY* regulates temperature stress response, we examined the expression levels of g*roEL* and *danJ*, which encode two heat shock proteins in *S. aureus*. In the absence of *codY*, the expressions of g*roEL* and *danJ* were significantly down-regulated (Figure 3E), indicating that *codY* possibly regulates the expression of g*roEL* and *danJ* in *S. aureus* to adapt to harsh temperatures.

Taken together, CodY may be a central temperature stress regulator that responds to environmental temperature changes by regulating the expression of heat shock proteins. In addition, CodY is required for *S. aureus* survival under high-temperature conditions. When codY is mutated, *S. aureus* barely survives at high temperatures.

### 3.4. CodY Is Involved in Regulating the Oxidative Stress Response of RMSA24

Hydrogen peroxide is a commonly used bacterial microbicide with a strong bactericidal effect on a variety of microorganisms [52]. *S. aureus* was treated with 5% hydrogen peroxide, and the survival rates were calculated. As shown in Figure 4A, the survival rates of WT and C-*codY* were higher than those of M-*codY*. These differences in survival rate became more significant as the duration of H₂O₂ treatment increased. When the treatment time was prolonged, the survival rate of the CodY mutant decreased dramatically (Figure 4B). This suggests that the mutation of *codY* leads to a more severe reduction in the tolerance of *S. aureus* to continuous oxidative stress. Taken together, CodY is an essential transcription factor for its oxidative stress response and positively regulates oxidative stress tolerance in *S. aureus*.

### 3.5. CodY Mutation Increases the Sensitivity of RMSA24 to Salt

*S. aureus* has a strong ability to tolerate salt stress, which enables it to survive under high-salt conditions [53]. In this study, we injected the bacterial suspension into TSB without or containing 10% NaCl; the bacterial suspension was then cultured for 10 h. In tryptone soybean broth medium without sodium chloride, the growth rates of the three strains tended to be consistent (Figure 5A). However, the growth rate of M-*codY* was reduced compared to WT and C-*codY* in TSB under high salt concentrations (Figure 5B). These results suggested that *codY* positively regulates the salt stress tolerance of *S. aureus* RMSA24.

## 4. Discussion

CodY is a transcription factor found in bacteria that controls the expression of a wide range of genes involved in amino acid metabolism, carbon metabolism, and virulence [39,54]. Some reports have shown that CodY might also be involved in biofilm formation, sporulation, and pathogenicity in certain bacterial species [55]. However, the function of CodY in environmental tolerance has not been reported. In this paper, we reported the function of CodY in environmental stress tolerance in *S. aureus* for the first time and explored its potential mechanistic response to temperature stress. The mutation of *codY* resulted in a significant reduction in the tolerance to desiccation and oxidative, salt, and harsh temperature stresses. Meanwhile, the expressions of the heat shock proteins GroEL and DnaJ were significantly reduced in the absence of *codY*. The above results suggest that CodY is an essential transcription factor involved in environmental tolerance in *S. aureus*.

Milk and dairy products generally undergo high- and low-temperature treatment, drying, and high oxidation during processing and storage [56]. Although these treatments can effectively control the growth of pathogenic bacteria and even inactivate them, there are numerous reports of infection resulting from *S. aureus* in milk or dairy products [57]. Due to the adaptability and stress resistance of *S. aureus*, it can survive in harsh environmental conditions. The production of heat shock proteins allows *S. aureus* to rapidly respond to temperature changes and helps the intracellular proteins fold and assemble properly [58]. The special composition of membrane lipids gives *S. aureus* resistance to dry environments. In-membrane cardiolipin may play a role in both structural stability and drying stress response [59]. Some antioxidant enzymes, such as antioxidant dismutase and catalase, can help *S.aureus* effectively resist oxidative stress in the environment, and some small molecules, such as bacillithiol, coenzyme A, and staphyloxanthin, are also involved in the oxidative stress response [60]. As a traditional food processing method, pickling is widely used in meat processing [6]. *S. aureus* can resist the stress caused by a high concentration of sodium ions in the environment through high levels of potassium ions in the cytoplasm, and this mechanism endows it with a strong tolerance to high concentrations of salt so that it can survive in pickled foods [61].

The resistance of *S. aureus* to stress is dependent on transcription factor regulation. Several transcriptional regulators are indispensable for environmental tolerance in *S. aureus*. The transcription factor SarA plays an integral role in regulating the environmental stress response of *S. aureus* [62]. SarA regulates the expression of genes that are involved in the oxidative stress response, antibiotic resistance, and biofilm formation [63,64]. Sigma factor B (sigB) is an important transcriptional regulator that influences the response of bacteria to environmental stress by regulating the expression of multiple genes [22,23,24]. In foodborne *S. aureus* RMSA24, the deletion of *sigB* leads to a significant increase in sensitivity to heat, drying, and oxidative stress [28]. Research on CodY has focused mainly on its regulatory role in metabolism and virulence. In *B. subtilis*, CodY is mainly involved in the regulation of carbon and nitrogen metabolic pathways and is critical for the transition between exponential and stationary phases of growth [65]. In *S. aureus*, CodY regulates the production of virulence factors and is essential for survival in the host environment [54]. In this study, we found that the absence of *codY* markedly increased the tolerance of RMSA24 to drying, oxidative, and salt stresses by comparing the survival rate after treatment. This suggests that codY affects metabolism and virulence while also acting as an important regulator of the environmental stress response.

GroEL and DanJ are important proteins involved in the heat stress response in *S. aureus* [66]. Under the conditions of heat stress, GroEL and DanJ expression levels are significantly increased, thereby enhancing the ability of cells to cope with temperature stress [45]. Some studies have found that the deletion or abnormal expression of GroEL and DanJ may increase the sensitivity of *E. coli* to high-temperature stress and other stress environments and even cause cell death [67,68]. In this paper, we found that the mutation of *codY* leads to a decreased tolerance to high-temperature stress and an increased tolerance to low-temperature stress in *S. aureus*. We examined the expression of *groEL* and *danJ* via RT-qPCR and found that the expressions of both genes were significantly down-regulated when *codY* mutated. These results suggest that CodY may indirectly affect the temperature stress response of *S. aureus* by regulating the expression of *groEL* and *danJ*. However, the complex regulatory mechanism requires further study.

There are many studies on the environmental stress response of *S. aureus*, but many aspects are still not well understood. For example, how *S. aureus* receives and responds to environmental stimuli and the transcription factors that play regulatory roles in this process are still not fully understood. For this paper, we studied the role of codY in environmental tolerance. However, how codY receives environmental stimulation and its specific regulatory mechanism require further study.

## 5. Conclusions

In this paper, for the first time in the literature, we described the role of the transcriptional regulator CodY in environmental tolerance and provided a preliminary insight into the mechanism by which CodY regulates the response to temperature stress in foodborne *S. aureus* RMSA24. We found that *S. aureus* with *codY* mutation showed both a decreased tolerance to desiccation and high-temperature, oxidation, and salt stresses and an increased tolerance to low-temperature stress. RT-qPCR analysis showed that the expressions of heat shock protein genes *groEL* and *danJ* were obviously down-regulated in the *codY* mutant strain, suggesting that CodY may indirectly regulate the heat stress response of *S. aureus* by affecting the expression of these proteins. This paper provides a new target gene for revealing the tolerance mechanism of *S. aureus* to the environment and controlling the contamination and harm caused by milk-derived *S. aureus*.

## Figures and Tables

**Figure 1 foods-12-03166-f001:**
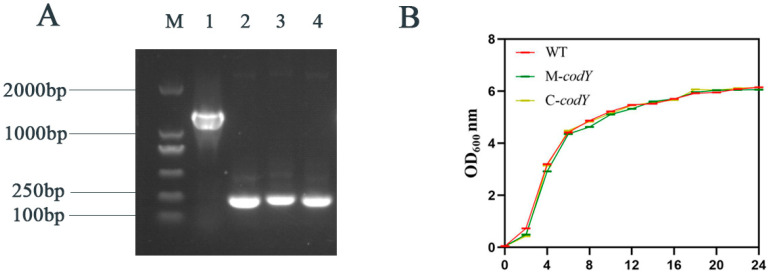
(**A**) Verification of the mutant strains. M: DNA marker; lane 1: WT; lane 2: M-*codY*; lane 3; C-*codY*; lane 4: pLI50. (**B**) Growth curves of WT, M-*codY*, and C-*codY*.

**Figure 2 foods-12-03166-f002:**
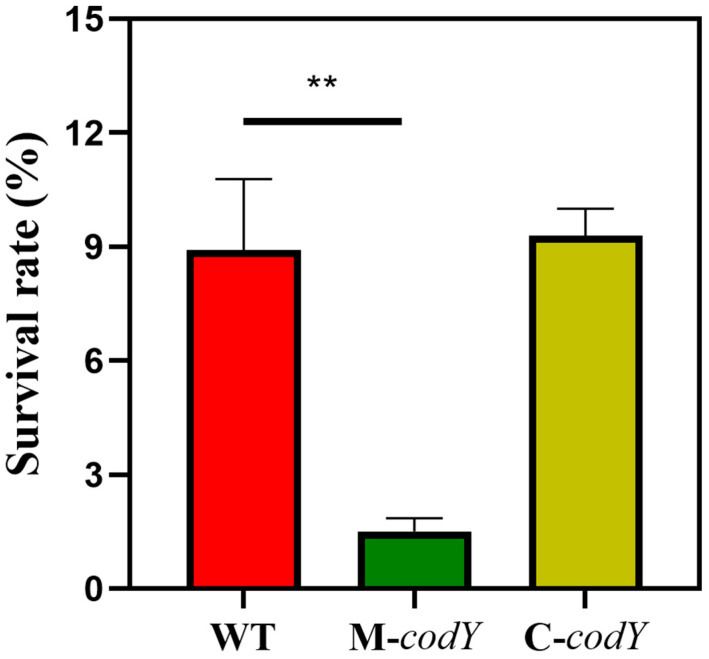
The survival rate of mutant strains after 4 d of drying stress. Statistically significant differences calculated by the unpaired two-tailed Student’s *t*-test are indicated: ** *p* < 0.01.

**Figure 3 foods-12-03166-f003:**
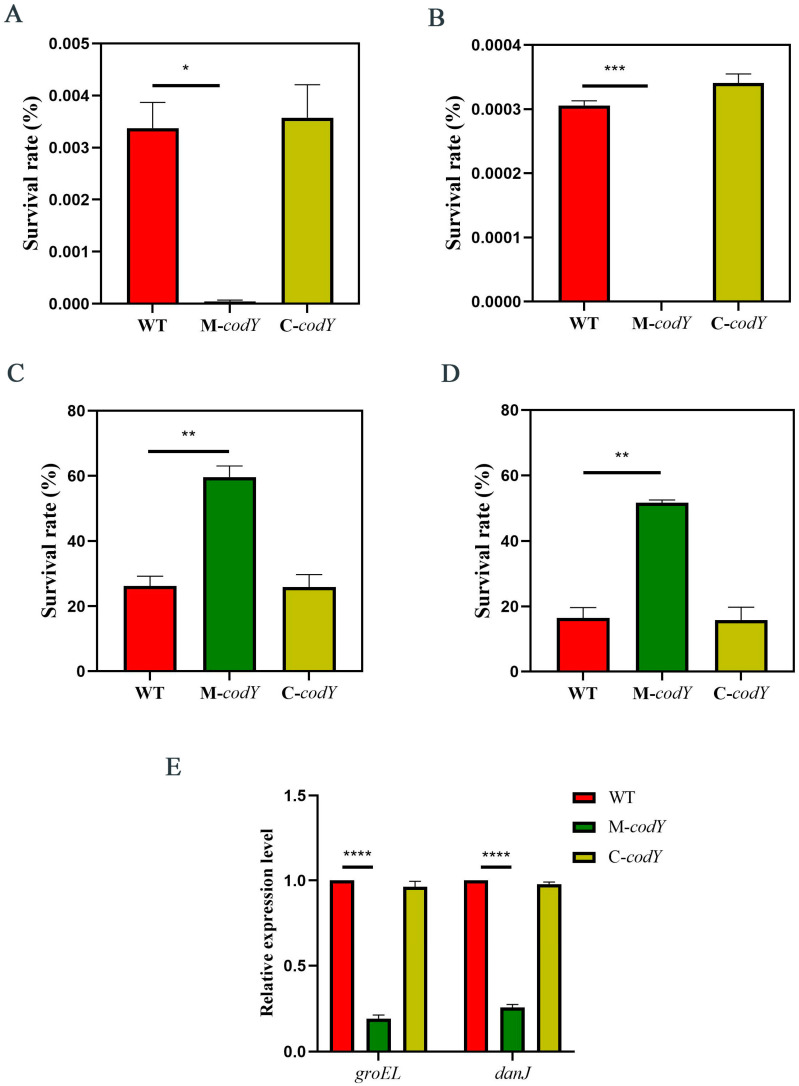
The survival rate after (**A**) 30 min or (**B**) 60 min of high-temperature stress (63 °C). The survival rate after (**C**) −20 °C or (**D**) −80 °C treated for 2 h. (**E**) Relative expression of *groEL* and *danJ*. The “*” indicates significant difference *p* values between groups. Statistically significant differences calculated by the unpaired two-tailed Student’s *t*-test are indicated: ** *p* < 0.01, *** *p* < 0.001, **** *p* < 0.0001.

**Figure 4 foods-12-03166-f004:**
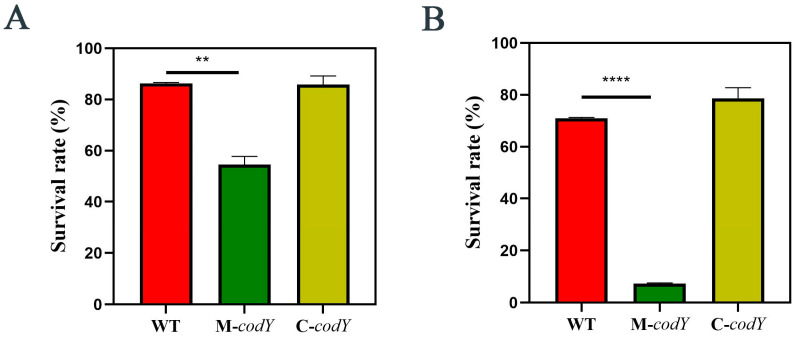
The survival rate after (**A**) 30 min or (**B**) 60 min of hydrogen peroxide treatment. Statistically significant differences calculated by the unpaired two-tailed Student’s *t*-test are indicated: ** *p* < 0.01; **** *p* < 0.0001.

**Figure 5 foods-12-03166-f005:**
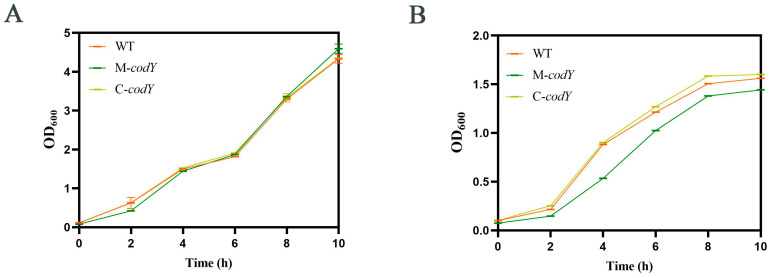
The growth curves of the three strains in TSB (**A**) without or (**B**) containing 10% sodium chloride.

**Table 1 foods-12-03166-t001:** Strains and plasmids used in this study.

Strain or Plasmid	Relevant Genotype	Reference or Source
Strains		
WT	RMSA24 with pLI50	Laboratory stock
*S. aureus* RN4220	8325-4, restriction-negative strain	NARSA
M-*codY*	RMSA24 *codY*-inserted mutant	This work
C-*codY*	RMSA24 *codY*-inserted mutant with pLI50-*cody*	This work
DH5α	Clone host strain, *supE44 ΔlacU169(φ80 lacZ*Δ*M15) hsdR17 recA1 endA1 gyrA96 thi-1 relA1*	Invitrogen
Plasmids		
pLI50	Shuttle vector, Ap^r^, Cm^r^	[49]
pLI50-*codY*	pLI50 with *codY*	This work

**Table 2 foods-12-03166-t002:** Primers used in this study.

Primers Name	Oligonucleotide (5′–3′)	Source
*codY*-F	TCTAGAGTCGACCTGCAGGCATGCAAGTGGTCAAGATGTCTCAAGAC	This work
***codY*-R**	TTATGCCTAAAAACCTACAGAAGCTTGTCCCAGACTCATCGACTTA	This work
check-pLI50-f	CCTGACGTCTAAGAAACCAT	This work
check-pLI50-r	CGATAACCACATAACAGTCA	This work
hu-F	AAAAAGAAGCTGGTTCAGCAGTAG	This work
hu-R	TTTACGTGCAGCACGTTCAC	This work
dnaJ-F	AGGATTCAATGGCTCTG	This work
dnaJ-R	TGTACCAAATACCGCTTC	This work
groEL-F	GTAGGTGCGATTTCAGC	This work
groEL-R	AAATGTATGGGCGTTCT	This work

## Data Availability

The data presented in this study are available upon request from the corresponding author.

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
