# Peer review of "CodY: An Essential Transcriptional Regulator Involved in Environmental Stress Tolerance in Foodborne Staphylococcus aureus RMSA24"

_foods, 2023, doi:10.3390/foods12173166_

Round 1
Reviewer 1 Report
Comments and Suggestions for Authors
Author Response
Reviewer #1:
- Lines 29 Gram
Reply: Thanks for your comments. We apologized for the irregularity. This irregularity has been corrected. (Lines 29)
- Lines 31 S. aureus in the remaining text
Reply: Thank you for your detailed and pertinent comments. The “Staphylococcus aureus” in the remaining text has been changed to " S. aureus ". (Lines 31, Lines 33, Lines 66, Lines 71, Lines 132, Lines 193, Lines 198, Lines 214, Lines 254, Lines 276)
- Lines 58 study
Reply: Thank you for your detailed and pertinent comments. We apologized for the irregularity. This irregularity has been corrected. (Lines 65)
- Lines 58 explain briefly how?
Reply: Thank you for your detailed and pertinent comments. This study found the function of CodY in environmental tolerance, and provided a theoretical basis for revealing the mechanism of stress resistance of Staphylococcus aureus. Understanding the stress resistance mechanism of Staphylococcus aureus can help us improve food processing technology to reduce the residue of Staphylococcus aureus in food.
- Lines 280 and Lines 292
Reply: Thanks for your comments. We have modified the format of the references, and the format of all the references has been unified.

Reviewer 2 Report
Comments and Suggestions for Authors
In the manuscript “CodY, an essential transcriptional regulator, is involved in environmental stress tolerance in foodborne Staphylococcus aureus”, the authors often refer to a their previous study and this makes the text approximate especially in the Materials and Methods section;
Line 13: usually causes diarrhea - usually causes intoxication with vomiting;
Line 55: (harsh temperature, drying, salt and H2O2) - generally H2O2 is not used in food manufacturing processes, but as a disinfectant; the authors should explain why they tested the effect of H2O2;
Line 83: All the strains - which?;
Line 92: The cultures of three strains were grown at 58℃ - why was the temperature of 58 °C chosen? in heat treatment processes it is generally at least 72 °C;
Line 99: The bacterial suspensions were inoculated into tryptone soybean broth at a ratio of 1:20 and incubated at 10℃ - why 10°C? generally food refrigeration is <5°C;
Line 105: plate and incubated under desiccation conditions - what were the parameters?
Line 111: or without 5% H2O2 for 30 or 60 minutes - 30 e 60 minutes? clarify better;
Line 151: Drying is a common sterilization method in dairy processing - it is a method of stabilization and conservation, not sterilization;
Line 216: This indicates milk-derived S. aureus was more tolerant to these stresses, which may lead to the persistent risk of food poisoning caused by S. aureus. - I think this sentence can't be supported by the results of this study;
Finally, I think the authors should strongly review the manuscript.
Author Response
Reviewer #2
- In the manuscript “CodY, an essential transcriptional regulator, is involved in environmental stress tolerance in foodborne Staphylococcus aureus”, the authors often refer to their previous study and this makes the text approximate especially in the Materials and Methods section
Reply: Thanks for your comments. Our laboratory has conducted in-depth research on the stress resistance of foodborne Staphylococcus aureus, and the methods involved are also established for the first time by us. Therefore, in terms of materials and methods, there are similarities with the previous published papers, and we have tried to avoid similarities in the statement, but there is no way to avoid some necessary vocabulary.
- Line 13: usually causes diarrhea - usually causes intoxication with vomiting
Reply: Thanks for your comments. According to your suggestion, we have replaced " usually causes diarrhea " with " usually causes intoxication with vomiting ". (Lines 14)
- Line 55: (harsh temperature, drying, salt and H2O2) - generally H2O2 is not used in food manufacturing processes, but as a disinfectant; the authors should explain why they tested the effect of H2O2
Reply: Thanks for your comments. Sodium hypochlorite, as a commonly used food disinfectant, kills microorganisms in the same way as hydrogen peroxide, using their strong oxidation. Therefore, we used hydrogen peroxide to expose Staphylococcus aureus to similar oxidation conditions.
- Line 83: All the strains - which?
Reply: Thanks for your comments. “All the strains” include WT, M-codY and C-codY.
- Line 92: The cultures of three strains were grown at 58℃ - why was the temperature of 58 °C chosen? in heat treatment processes it is generally at least 72 °C
Reply: Thanks for your comments. Our heating treatment conditions are based on pasteurization. Pasteurization is generally to heat the milk to 60℃ and keep it for 30 minutes.
6.Line 99: The bacterial suspensions were inoculated into tryptone soybean broth at a ratio of 1:20 and incubated at 10℃ - why 10°C? generally food refrigeration is <5°C
Reply: Thanks for your comments. We apologize for our inaccurate description. In our experiment, there is no treatment group of 10℃. We have deleted it. (Lines 107)
- Line 105: plate and incubated under desiccation conditions - what were the parameters?
Reply: Thanks for your comments. The parameters of desiccation conditions are as follows: humidity: 25% ± 5%; Water activity: 0.38075 ± 0.01176. We have added the relevant parameters of drying conditions in this paper. (Lines 112)
- Line 111: or without 5% H2O2 for 30 or 60 minutes - 30 e 60 minutes? clarify better
Reply: Thanks for your comments. We apologize for the unclear statements and have clarify relevant parts. (Lines 119-121)
- Line 151: Drying is a common sterilization method in dairy processing - it is a method of stabilization and conservation, not sterilization.
Reply: Thanks for your comments. According to your suggestion, the “sterilization” has been replaced by “stabilization and conservation”. (Lines 160)
- Line 216: This indicates milk-derived S. aureus was more tolerant to these stresses, which may lead to the persistent risk of food poisoning caused by S. aureus. - I think this sentence can't be supported by the results of this study
Reply: Thanks for your comments. Our study really does not support this sentence, we're just citing other studies that have found that some Staphylococcus aureus isolates from food are very tolerant to the environment. Our study was designed to explore the effects of codY on environmental tolerance of food-borne Staphylococcus aureus. In addition, due to the short length of the paper, we have revised the discussion section, and in order to make the preceding and following texts logically rigorous, we have removed this sentence. (Lines 223)
- Finally, I think the authors should strongly review the manuscript.
Reply: Thanks for your suggestion. We have carefully revised the manuscript in accordance with the comments of all reviewers.

Reviewer 3 Report
Comments and Suggestions for Authors
The article is well written and the research project was well set up and well carried out. Furthermore, the article takes into consideration aspects of considerable interest for the study of the behavior of S. aureus in food for humans. I really enjoyed the contents and I congratulate the authors.
I have NO particular observations to make in this regard, I have made small corrections of typos in the pdf format which I attach with my comments in yellow.

Author Response
Reviewer #3
- Lines 11: I guess it's "institute" ...
Reply: Thanks for your suggestion. We apologized for the irregularity. This irregularity has been corrected. (Lines 11)
- Lines 29: Gram in capital letter
Reply: Thank you for your detailed and pertinent comments. We apologized for the irregularity. We have replaced " gram " with " Gram ". (Lines 29)
- Lines 58: study
Reply: Thank you for your detailed and pertinent comments. We apologized for the irregularity. This irregularity has been revised. (Lines 65)
- Lines 105: I think it is useful for the reader to better specify which conditions have been set to create this desiccation (temperature, air humidity, etc.)
Reply: Thanks for your suggestion. According to your suggestion, we have added the relevant parameters of drying conditions in this paper. (Lines 112)
- Lines 159: in italic, please.
Reply: Thank you for your detailed and pertinent comments. This irregularity has been corrected. (Lines 168)
- Lines 216: This indicates that milk-derived ..
Reply: Thank you for your detailed and pertinent comments. Due to the short length of the paper, we have revised the discussion section, and in order to make the preceding and following texts logically rigorous, we have removed this sentence. (Lines 223)

Round 2
Reviewer 2 Report
Comments and Suggestions for Authors
I thank the authors for their revisions, but in my opinion some critical points remain
Line 52: Streptococcus thermophilus
Line 101: The cultures of three strains were grown at 58℃ for thirty minutes or sixty minutes, respectively.
Reply: Our heating treatment conditions are based on pasteurization. Pasteurization is generally to heat the milk to 60℃ and keep it for 30 minutes
If heating treatment conditions are based on pasteurization, why did the authors use 58 °C?; generally the milk is pasteurized at 72 °C/ 15 seconds (High Temperature Short Time); when transformed into cheese it can be pasteurized at 63 °C/30 minutes (Low Temperature Long Time). The title of the manuscript speaks of foodborne Staphylococcus aureus; What foods are processed at 58°C?
In my opinion the authors should refer the results only to S. aureus strain RMSA24 and to the experimental conditions used ; for this reason the title should be changed to “CodY, an essential transcriptional regulator, is involved in environmental stress tolerance in Staphylococcus aureus strain RMSA24”
Author Response
Line 52: Streptococcus thermophilus
Reply: Thank you for your detailed and pertinent comments. This irregularity has been corrected. (Lines 87-88)
If heating treatment conditions are based on pasteurization, why did the authors use 58 °C?; generally the milk is pasteurized at 72 °C/ 15 seconds (High Temperature Short Time); when transformed into cheese it can be pasteurized at 63 °C/30 minutes (Low Temperature Long Time). The title of the manuscript speaks of foodborne Staphylococcus aureus; What foods are processed at 58°C?
Reply:Thank you for your suggestions. According to your suggestions, we have redone the high temperature stress experiment and changed the treatment temperature to 63 ° C. (Lines 137, Lines 206, Lines 225 and Lines 226)
In my opinion the authors should refer the results only to S. aureus strain RMSA24 and to the experimental conditions used; for this reason the title should be changed to “CodY, an essential transcriptional regulator, is involved in environmental stress tolerance in Staphylococcus aureus strain RMSA24”
Reply:Thank you for your detailed and pertinent comments. According to your suggestion, we have revised the title of this paper. (Lines 4).
